# Small Non-Coding RNAs at the Crossroads of Regulatory Pathways Controlling Somatic Embryogenesis in Seed Plants

**DOI:** 10.3390/plants10030504

**Published:** 2021-03-09

**Authors:** Ana Alves, Daniela Cordeiro, Sandra Correia, Célia Miguel

**Affiliations:** 1BioISI—Biosystems & Integrative Sciences Institute, Faculty of Sciences, University of Lisboa, 1749-016 Lisboa, Portugal; acalves@fc.ul.pt; 2Instituto de Tecnologia Química e Biológica António Xavier, Universidade Nova de Lisboa, 2780-157 Oeiras, Portugal; 3Centre for Functional Ecology, Department of Life Sciences, University of Coimbra, Calçada Martim de Freitas, 3000-456 Coimbra, Portugal; danielacordeiro@outlook.pt (D.C.); sandraimc@uc.pt (S.C.); 4iBET, Instituto de Biologia Experimental e Tecnológica, Apartado 12, 2781-901 Oeiras, Portugal

**Keywords:** angiosperms, auxin-responsive genes, early embryogenesis, somatic embryo maturation, gymnosperms, miRNAs

## Abstract

Small non-coding RNAs (sncRNAs) are molecules with important regulatory functions during development and environmental responses across all groups of terrestrial plants. In seed plants, the development of a mature embryo from the zygote follows a synchronized cell division sequence, and growth and differentiation events regulated by highly regulated gene expression. However, given the distinct features of the initial stages of embryogenesis in gymnosperms and angiosperms, it is relevant to investigate to what extent such differences emerge from differential regulation mediated by sncRNAs. Within these, the microRNAs (miRNAs) are the best characterized class, and while many miRNAs are conserved and significantly represented across angiosperms and other seed plants during embryogenesis, some miRNA families are specific to some plant lineages. Being a model to study zygotic embryogenesis and a relevant biotechnological tool, we systematized the current knowledge on the presence and characterization of miRNAs in somatic embryogenesis (SE) of seed plants, pinpointing the miRNAs that have been reported to be associated with SE in angiosperm and gymnosperm species. We start by conducting an overview of sncRNA expression profiles in the embryonic tissues of seed plants. We then highlight the miRNAs described as being involved in the different stages of the SE process, from its induction to the full maturation of the somatic embryos, adding references to zygotic embryogenesis when relevant, as a contribution towards a better understanding of miRNA-mediated regulation of SE.

## 1. Introduction

Small non-coding RNAs (sncRNAs) are usually 20–24 nt long RNA molecules with important regulatory functions during development and environmental responses across all groups of land plants, from bryophytes like *Physcomitrella patens* [1] and *Marchantia polymorpha* [2], to the gymnosperms [3] and angiosperms [4]. These small molecules are classified into several classes according to their biogenesis and roles [5], the micro RNAs (miRNAs) being the best characterized group of sncRNAs, and the short-interfering RNAs (siRNAs) the other broad category of plant sncRNAs. miRNAs are typically molecules with 21–22 nt produced from a *MIR* gene. Commonly, these tiny molecules act at the post-transcriptional level based on nearly perfect sequence complementary recognition of their target mRNAs, leading to the cleavage or translation inhibition of the target. The most abundant subclasses of siRNAs, with 24-nt or 21–22-nt, can either lead to transcriptional or post-transcriptional gene silencing depending on if they derive from transposons or repeats (heterochromatic siRNAs), or if they are triggered by specific miRNAs (trans-acting siRNAs) [6]. The involvement of these classes of molecules in several plant developmental processes, including embryogenesis, and in response to environmental conditions has been often reported and reviewed [4,7,8,9].

Somatic embryogenesis (SE) refers to the process whereby somatic cells can be induced to undergo a series of developmental stages that mirror the development of a zygotic embryo within the seed and has been widely used as an experimental model to study zygotic embryogenesis (ZE). In *Arabidopsis thaliana* (Arabidopsis), the distinct stages of embryo development are characterized in detail and include the zygotic, globular, heart, torpedo and cotyledonary (mature) embryo stages [10]. At the zygotic stage, asymmetric cell division occurs, originating a small apical cell from which most parts of the mature embryo will form, and a basal cell whose divisions later originate the distal part of the root apex and the suspensor. The globular embryo results from a series of cell divisions, and at the 16-celled proembryo it is possible to identify a superficial cell layer, the protoderm, that will become the epidermal tissue. Polar auxin transport mediated by the PIN proteins is critical for establishing the distal part of the root apical meristem and later on for the formation of the two cotyledon primordia at the heart stage. At the latest stages of embryogenesis, the torpedo and mature embryo, the basic plan of the future plant is formed, bearing the root apical meristem at one end and the shoot apical meristem localized between the two cotyledons at the other end.

Given the distinct features found right from the very initial stages of embryogenesis in gymnosperms and angiosperms, it is relevant to investigate to what extent such differences emerge from differential post-transcriptional regulation mediated by sncRNAs. As an example, the critical event in early embryogenesis of angiosperms whereby the asymmetric cell division of the zygote originates the apical and the basal cell, which will follow different cell-fate specification pathways, does not occur in gymnosperms. In these species, the zygote goes through several rounds of nuclear duplication without cytokinesis followed by cellularization to form two tiers of cells and a proembryo with eight cells [11]. Recent work uncovering cell lineage-specific transcriptome in Arabidopsis shows distinct molecular pathways at the apical and basal cell [12] following the first cell division, a stage that has no morphological equivalent in gymnosperm embryogenesis.

Another study with Arabidopsis mutants in genes required for miRNA biogenesis (*dcl1*, *se-1* and *hyl1-2*) provided evidence that miRNAs are required at the zygote stage [13]. Some of these mutants showed loss of zygote polarity, presenting a more symmetric pattern of the first cell division and subsequent horizontal division of the apical cell, instead of vertical division. These observations lead to several pertinent questions that can start to be addressed by gathering the evidence available in several reports in the literature. For instance, it is important to determine if initial zygotic/somatic embryo development in gymnosperms is regulated by miRNAs, if the regulation is mediated by the same miRNA families in both gymnosperms and angiosperms and if the roles of these miRNAs are conserved, i.e., if their target transcripts show conserved functions.

The same kinds of questions apply to other stages of embryo development, where distinct features are evident in the two plant groups. In fact, in addition to the occurrence of cleavage polyembryony, a common phenomenon in gymnosperms during early embryogeny, the differentiation of multiple cotyledons, instead of only two cotyledons, around the shoot apical meristem (SAM) during late embryogeny, is also a highly distinctive characteristic that confers radial symmetry to the embryo instead of the bilateral symmetry observed in angiosperms [14]. Furthermore, it is also important to focus on the events that occur prior to embryo development, during the induction of SE, where a somatic cell changes its fate in order to become an embryo.

Here, we gather the current knowledge on the presence and characterization of sncRNAs, especially miRNAs, during embryogenesis, focusing on these two major groups of plants, the angiosperms and gymnosperms. Despite the aforementioned differences in embryo development, highlighting the similarities and the main differences already identified so far will help to address if regulation of SE by sncRNAs occurs in a similar way in these two groups. Within the gymnosperms, SE is especially relevant in conifers where it offers potential benefits in the clonal propagation of selected germplasm. However, one of the main obstacles to its industrial application has been the limited range of tissues from which SE can be induced. This fact can be hypothetically related to the repertoire of transcripts, coding and non-coding, present in the tissues at the time of excision (explant preparation). In the sections below, we start by providing a broad overview of the sncRNA landscape in embryo tissues, followed by a more detailed description of already characterized miRNAs related to several stages of embryogenesis, from the induction phase to later stages of embryo development, focusing on SE.

## 2. Small RNA Expression Profiles in Embryonic Tissues

Seed tissues, including the embryos, show characteristic expression profiles. In Arabidopsis, a recent study highlighted the uniqueness of embryo transcriptomes when compared to other post-embryonic tissues [15]. Given the role of sncRNAs in the control of gene expression, it is not surprising that sncRNA transcriptomes of embryos have revealed characteristic features underling some of the regulatory pathways that operate at specific developmental stages. While diverse sncRNA profiles have been detected in seed tissues, based mostly on available data from angiosperms, there seems to be a general tendency for a higher abundance and diversity of 24-nt sncRNAs regardless of the developmental stage of the seed [16]. In some gymnosperm species, like maritime pine (*Pinus pinaster*), 24-nt sncRNAs were also highly expressed in embryos when compared to vegetative tissues [3,17] and more abundant in somatic embryos than in zygotic embryos at equivalent stages of development. In *Picea glauca*, another gymnosperm species, both 21-nt and 24-nt sncRNAs were abundant in embryos [18], this profile being in contrast to that of buds [19], confirming that the 24-nt sncRNAs are mostly present in reproductive tissues. It has been proposed that such abundance is associated with epigenetic safeguard mechanisms for controlling the expression of transposons and heterochromatic repeats during embryo development [16,20]. A recent study [21] showed that siRNAs generated from transposable elements (TEs), highly numerous in Arabidopsis embryos and typically 24-nt long, originate both from euchromatic TEs, being required not only during embryogenesis as a way of silencing TEs, but also from heterochromatic TEs specifically required during embryogenesis to help establish TE methylation *de novo*, which is then maintained during post-embryogenesis independently of siRNAs.

Despite the recognized importance of siRNAs during embryogenesis and its apparent increased abundance in somatic embryos, most probably as a response to artificially-provided environmental conditions, these sequences remain mostly uncharacterized, with most of the attention being focused on miRNAs, especially those that are conserved among angiosperms.

Although many miRNAs are conserved and significantly represented across angiosperms and other seed plants during development, including embryogenesis, some miRNA families exist that are specific to some plant lineages (Table 1).

Some of these specific or non-conserved miRNAs might play yet uncovered roles in embryo development. For example, in the conifer *P. pinaster*, in addition to miRNA families conserved across land plants, eleven miRNA families, such as miR1316 and miR3699, were detected in embryos and megagametophytes that are conifer-specific. Among these, miR946, miR947, miR950, miR951, miR1311, miR1312, miR1313 and miR3701 were also found in the transition from dormancy to germination of Japanese larch (*Larix leptolepis* (syn. *L. kaempferi*)) embryos [22], and miR1315 was identified in embryogenic tissues of *Picea balfouriana* [23]. Other conifer-specific miRNAs are miR3702 and miR3704, which were identified in SE-related tissues of *L. leptolepis* [24]. In turn, miR1314 family is gymnosperm-specific, since in addition to being identified in *P. pinaster* embryos and megagametophytes [16] and *L. leptolepis* embryos [22], it was found in *Ginkgo biloba* leaves [25]. Similar observations have been documented for other species [18]. For instance, some Poaceae-specific miRNAs, such as miR531, miR1139, miR1878 and miR5049, were reported as being involved in the embryogenic callus development [26], as well as the angiosperm-specific miR444 that has revealed a differential accumulation during SE induction in maize (*Zea mays*) [27].

## 3. Induction and Early Somatic Embryogenesis

The transition from somatic cells to embryonic tissues is a widely reviewed process due to its biotechnological importance, from the micropropagation of superior genotypes to the production of transgenic plants. While zygotic embryo development starts with the formation of the zygote following fertilization, somatic cells acquire embryogenic competence as a result of forced chemical and physical stimuli that induce the reprogramming of gene expression patterns [35,36,37].

The plant embryogenesis induction mechanisms are complex but quite similar among different plant species: (i) it is required that the explant cells have the potential to express totipotency, (ii) these cells must be competent to respond to exogenous stimuli and (iii) the competent cells, induced by specific stimuli, become committed to an embryogenic fate. Embryogenesis commitment is preceded by a dedifferentiation/transdifferentiation stage [38], in which differentiated cells from the explant lose their specificity and acquire a meristematic-like behavior [39]. Then, depending on the culture conditions provided, those cells can start a new developmental fate and regenerate embryos and eventually a complete plant. Thereby, the SE starts with embryonic induction in which the expression of a set of genes is promoted by specific stimuli, most commonly the application of exogenous plant growth regulators (PGRs) or stress conditions, to form the embryogenic cells. Several studies have focused on the earliest stages of SE induction [40,41,42,43,44,45], trying to clarify the endogenous and exogenous factors associated with the embryonic switch in order to better understand the molecular mechanisms involved in the developmental cell plasticity and improve plant regeneration systems. However, SE can be initiated in different ways depending on the species, the explant type and the provided stimuli. Somatic embryos can initiate through direct and indirect embryogenesis, and from the organization of groups of cells relying on gradients of auxin and cytokinin and simultaneous establishment of meristem organizing centers [38]. Therefore, it should be pointed out that the underlying molecular mechanisms involved, including those associated with miRNA functions, cannot be generalized and may differ depending on the developmental pathway followed.

The crucial role of miRNAs in a great number of developmental processes in plants has been reported, including ZE, in which miRNAs were identified to be fundamental to the proper patterning and morphology of the embryos [46]. Taking into account the substantial impact of miRNAs in ZE regulation and their reported roles in plant cell response to hormones or stress conditions, it is expected that these molecules are also crucial in the regulation of SE, from its induction phase up to the development of a fully mature embryo. In fact, the expression of miRNAs during SE induction was reported for several species, within both angiosperms and gymnosperms; however, a comprehensive functional characterization of the specific miRNAs identified is still limited. Interestingly, it has been reported that *MIR* gene promoters are enriched in regulatory sequences that control hormone and stress responses [47,48]. A recent study focused on the regulation of Arabidopsis SE [49] reported a vast number of active *MIR* genes (98%), some of which were specifically expressed in the induction phase. From these, a high number of differentially expressed *MIR* genes during early SE (56%) and advanced SE (58%) stages were identified, but their expression patterns differed sharply between the two SE phases. During early SE induction (0–5 days), a majority of *MIR* genes were downregulated, and a large part of them was found to be highly repressed. In opposition to early SE induction, most *MIR* genes were predominantly upregulated in advanced SE. These observations support previous evidence of the miRNA contribution to cellular differentiation during embryonic development through the regulation of transcription factor (TF) genes in both SE and ZE [50]. In fact, the differential expression of a substantial number of miRNAs [49] is accompanied by an extensive modulation of TF genes in Arabidopsis embryogenic cultures [49,51]. Furthermore, *dcl1* mutants, harboring a deficiency in the *DICER-LIKE1* (*DCL1*) gene with a crucial role in miRNA biogenesis, were reported as unable for SE induction [52].

Moreover, evidence of the involvement of miRNAs in in vivo asexual embryogenesis, a good model to understand in vitro somatic embryogenesis, has emerged. While studying nucellar embryony (initiation of asexual embryos directly from nucellar cells surrounding the embryo sac) in citrus [53], the authors of the study found that from the approximately 150 miRNAs, including ~90 conserved and ~60 novel miRNAs, expressed in the ovules of both poly and monoembryonic ovules, two of them were differentially expressed. The novel miRNA named miRN23-5p was repressed in the polyembryonic ovules, and its target showed a reciprocal expression pattern, suggesting putative involvement in the process.

### 3.1. Plant Growth Regulators and Stress Signaling Associated miRNAs

A deep knowledge of the physiological and molecular mechanisms underlying induction of SE is fundamental for its manipulation. There are several factors commonly reported as promoting the induction of SE, including culture conditions such as medium composition, high concentrations of PGRs and the wounding of the explant. The type of explant, the genotype, the cellular density and the explant age are additional factors that influence embryogenic potential acquisition [44].

Considering the numerous protocols found in the literature, SE can be induced from different explant types, like seedlings, petioles, leaves, roots, shoot meristems, seeds, cotyledons and zygotic embryos. Nevertheless, immature zygotic embryos are the most frequently used, allowing one to induce SE in species considered for many years to be recalcitrant, such as conifers [54] and many angiosperms, including Arabidopsis [40]. SE-responsive explants in many species seem to contain a higher indole-3-acetic acid (IAA) content than non-responsive explants, suggesting a positive correlation between explant-responsiveness and IAA content. However, in other species including conifers, the SE-recalcitrant tissues also presented high level of IAA, implying that the specific endogenous auxin content that presumably enables SE induction seems to be genotype- and tissue-dependent [45].

In combination with genotype and tissue specificity, the effect of different stimuli is very important in triggering the molecular mechanisms underpinning SE induction. Accordingly, in Arabidopsis, several genes that include many targets of miRNAs associated with embryogenic response (e.g., miR156, mir157, miR158, miR159, miR160, miR164, miR166, miR169, miR319, miR390, miR393, miR396, miR398) are annotated to the functional category response to stimuli, including those involved in plant responses to PGRs, especially auxin, abscisic acid (ABA), gibberellic acid (GA3) and ethylene, osmotic stress and radiation [49].

The miR169 is a stress-related candidate possibly involved in the regulation of SE, given its high expression in response to different stresses in distinct plants like Arabidopsis, tomato (*Solanum lycopersicum*) and rice (*Oryza sativa*) [55]. Consistently, in Arabidopsis this can be corroborated by the down-regulation of NF-YA gene family members (*NF-YA1*, *NF-YA8* and *NF-YA10*, encoding the HAP2-type TFs), established as the main targets of miR169 family [49,56,57,58], in the early stages of SE induction, simultaneously with the high expression of miR169h-n [49].

Similarly, the differential expression of miR319 during Arabidopsis SE suggests that this miRNA, involved in the control of the general plant stress-responses, also plays a role in SE induction by repressing the auxin response inhibitor *AtIAA3* and by indirectly interfering with auxin signaling [59]. During advanced stages of SE induction, with a large emphasis on somatic cell differentiation, miR319 was reported to control *TCP4* and *TCP10* (*TEOSINTE BRANCHED1/CYCLOIDEA/PROLIFERATING CELL FACTOR*) genes encoding TFs that are involved in the organ-specific regulation of cell growth and differentiation [49,60].

Additionally, miR398, with differential accumulation in SE, was reported to control plant responses to stress, contributing to SE induction via activation of a stress reaction [61]. Recently, this hypothesis was supported in Arabidopsis by evidence of down-regulated expression of miR398 in early SE followed by a significant up-regulation of the *CSD1* gene (*Cu/Zn SUPEROXIDASE DISMUTASE1*) that encodes an enzyme involved in the response to oxidative stress [49]. The downregulation of miR398 linked with the up regulation of the *CSD* genes was also reported in embryogenic cultures in other angiosperms like longan (*Dimocarpus longan*) [32] and conifers, like *L. leptolepis* [24].

The miR164, targeting genes of the NAC transcription factor family, including the *CUP-SHAPED COTYLEDON1* (*CUC1*) and *CUC2* genes, has previously been reported as required for normal embryonic development contributing to the separation of adjacent developing organs [62]. In addition to this role, miRNA164 has been suggested to have a regulatory role during SE induction in Arabidopsis [49]. While CUC1 and CUC2 are highly expressed, miR164 is downregulated during induction. However, it should not be excluded that other NAC TFs previously reported to be involved in auxin signaling [63] or stress responses [64,65] are targeted by members of the miR164 family during SE induction.

In what concerns miR159, it has been highlighted in SE induction both in Arabidopsis, showing a differential accumulation during SE induction [49] and in conifer species. In the conifer *L. leptolepis* [66], the maintenance of the embryogenic potential in proliferating embryogenic cultures has been reported as associated to the regulation of LaMYB33 transcript levels by its targeting miR159. This is an example of a miRNA that seems to be important not only in maintaining the ability for embryogenic potential but also in the somatic embryo maturation process, depending on its expression level. A low expression of miR159, and a corresponding increased expression of *LaMYB33*, was found in non-embryogenic cultures, which may be associated with ABA signaling commonly associated with embryo maturation (see Section 4). Additionally, in *P. balfouriana*, possible regulation of GAMYB-like gene expression by miR159 was associated to embryogenic ability through mediation of GA3 levels in cultures subject to different cytokinin treatments [23]. In the same study, the AP2 domain-containing transcription factor family, which includes, for example, BABY BOOM (BBM), with well-established roles in SE induction [67], has been implicated in miRNA-mediated regulation associated to embryogenic competence, namely involving miR1160.

Finally, it is important to mention a recent study in Arabidopsis [68] showing that the control of miRNA pathways can be mediated by histone acetylation during the embryogenic reprogramming. This involved the transcription factor AGAMOUS-LIKE 15 (AGL15), which is capable of regulating miR156 both by transcriptional activation of *MIR156* or containment of its levels through repression of the miRNA biogenesis genes *DCL1*, *SERRATE* and *HEN1*. In experiments involving the use of an inhibitor of the HDAC histone deacetylases, the authors showed that histone deacetylation is associated with the repression of miRNA processing, and this is mediated by AGL15 [68].

### 3.2. Auxins and miRNA–ARF Interactions

As mentioned above, auxin content has been pointed out as one of the most important factors required for the transition of a somatic cell into an embryo [45]. High concentrations of auxin are often related to SE stimulation in plants. Auxin signaling is activated by high levels of exogenously supplemented auxin to explant tissues. This network is crucial for the reprogramming of gene expression cascades during somatic cells to embryo transition [69]. The most used synthetic auxin for SE induction is 2,4-dichlorophenoxyacetic acid (2,4-D), structurally and functionally analogous to the natural auxin IAA, being applied in almost 78% of the published protocols, either alone or in combination with other PGRs [45].

Such as for the Arabidopsis model system, most frequently used for better understanding the regulation of SE, many differentially expressed miRNAs have also been identified in other embryogenic cultures of different gymnosperms [18], mono- and dicotyledonous plants [15,21,26,27]. From these studies, a large number of miRNAs has been associated with SE target genes related to auxin perception, signaling and biosynthesis, as presented in Figure 1.

The most represented auxin-related miRNAs in embryogenic cultures are miR165/166 and miR167, which have been found in all SE studies, miR160, miR164 and miR390, which have been identified in most SE transcriptomes and miR395, with expression in some of the embryogenic cultures. In addition to these widely conserved miRNAs, an angiosperm-specific miRNA, miR827, has been suggested to have a role in the embryogenic ability acquisition through the regulation of auxin metabolism in *D. longan* callus [30].

Given that several TF binding sites were identified in the *MIR* promoters, regulatory feedback loop mechanisms between TFs and *MIR* genes are expected to operate during SE [47]. In fact, Hewezi and Baum (2012) [81] found that in Arabidopsis, the *GROWTH RESPONSE FACTOR1* (*GRF1*) and *GRF3*, both targets of miR396, which were differentially expressed in SE, were also reported to repress the expression of miR396a and miR396b. Furthermore, *AUXIN RESPONSE FACTORS* (*ARFs*) might control the expression of their targeting miRNAs, MIR160, MIR167 and MIR390, during SE, on account of the presence of *AUXIN RESPONSE ELEMENTS* (*AREs*) detected in the promoters of these genes [49,82].

miR160 has been indicated as an important auxin-related miRNA targeting *ARF10*, *ARF16* and *ARF17* [42,83,84], and being involved as regulator of several developmental processes in Arabidopsis, including embryogenesis. As reported in the conifer *L. leptolepis* [24] and in *D. longan* [32,85], the downregulation of miR160 was also observed in Arabidopsis SE [42,49].

The association of miR165/166 with the control of auxin biosynthesis during SE has been reported, corroborating the enhanced embryogenic response of the adaxial vs. abaxial side of cotyledon explants in Arabidopsis immature zygotic embryos [86]. Such a response was promoted by the restriction of PHABULOSA/PHAVOLUTA (PHB/PHV) transcripts to the adaxial cotyledon side by the action of miR165/166, resulting in a side-specific auxin accumulation, caused by the stimulation of *LEAFY COTYLEDON2* (*LEC2*) expression mediated by PHB/PHV [42]. Additionally, ARF10 and ARF16 are referred as possible negative regulators of PHB/PHV since there is an increase of PHB transcript levels in *arf10arf16* and a decrease in ARF16, miR160b and miR160c. An interaction between miR160 and miR165/166 during the SE induction process was supported by an up-regulation of ARF10 and ARF16 transcript levels in *phb*, *phv*, *phb1-d* and short tandem target mimic (STTM) 165/166 lines [42]. The LEC1 and LEC2 targets diversity suggested that these proteins are important contributors for several auxin-related processes, such as auxin biosynthesis, and members of the Aux/IAA family were indicated as regulators of LEC1 and LEC2 during embryo development [87]. Despite this information, the exact nature of miR160 and miR165/166 pathways interaction needs to be enlightened in future studies.

In Arabidopsis, a significant increase in the expression of *ARF5*, *ARF6*, *ARF8*, *ARF10* and *ARF16* is observed during the ZE and SE processes [41]. In a recent study with cotton (*Gossypium hirsutum*) SE, an increase of *ARF6* and *ARF8* expression was reported using miR167 mimic transformed lines (MIM167) for miR167 downregulation, in comparison with the control. In the same study, different genes showed significant differences in expression in MIM167 transformed lines when compared to control, namely auxin-response *GRETCHEN HAGEN 3* (*GH3*), auxin transporter *AUX1*, auxin influx/efflux carrier *LAX3*, encoding an AUX1-like protein 3, and *PIN-FORMED1* (*PIN1*) and *PIN2* genes, suggesting that diminution of miR167 led to enhanced cellular auxin signaling. Other miRNAs associated with auxin signaling, such as miR160, miR393, miR166, miR156 and miR157 were differentially expressed in MIM167 lines, suggesting that the expression of these miRNAs is influenced by the diminution of miR167 [88].

In a recent study, it was found that *MIR167A* acts as a maternal gene for embryo development in Arabidopsis mainly through the targeted *ARF6* and *ARF8* genes [89]. Although it seems that miR167A is the key DCL1-processed miRNA involved in embryo development control from maternal sporophytic tissues, the embryogenesis and seed development defects in *mir167a* mutants are 100% penetrant while *dcl1* mutants were not, suggesting that other DCL proteins may also be involved in miRNA167 precursors biogenesis [89]. Another recent research reported an increased auxin biosynthesis in the integuments at early embryogenesis stages, this maternally produced auxin being required for embryo development [90]. A lack of miR167 in the integuments will lead to an expected ARF6 and ARF8 target transcript accumulation, eventually disrupting auxin signaling and consequently seed development. Apart from this influence from maternal tissues, *MIR167A* and *MIR167B* were also found expressed in globular embryos [13], which corroborates its regulatory role of the auxin response in embryos thus contributing to normal embryo development. In what concerns SE, the involvement of these miRNAs in embryo development has not yet been reported. However, low levels of miR167 expression in cotton embryogenic *calli* as compared with non-embryogenic have been associated to a higher magnitude of SE in vitro [80], which is consistent with the results obtained for miR167-mimic transgenic *calli* in a recent study [88].

Regarding miR390, it regulates auxin signaling by directing the production of trans-acting small interfering RNAs (tasiRNAs) that down-regulate the expression of *ARF2*-*4* genes [91,92]. Even though miR390 had a significantly modulated expression in SE of Arabidopsis and in other plant species such as *L. leptolepis* [24], *D. longan* [32] and *Citrus sinensis* [93], results in Arabidopsis suggest that the regulatory complex miR390-TAS3-ARFs seems to operate during early SE induction and the regulation of ARF2 and ARF3 mediated by miR390 contributes to auxin signaling during embryogenic transition in induced somatic cells [49]. Additionally, miR393, targeting the auxin receptors TIR1 and AFB2, contributes to embryogenesis transition modulating tissue sensitivity to auxin treatments. Corroborating this assumption, the authors reported a relation between the embryogenic response of explant tissue to the level of expressed miR393 and the concentration of 2,4-D used in the medium for SE induction [52].

## 4. Late Somatic Embryogenesis

Somatic embryo full development and maturation prior to germination is a crucial step to achieve plant formation through SE. As it would be expected, the regulatory networks involved in the acquisition of the features characteristic of a mature embryo are different from those operating in the initial phases of SE induction and morphogenic embryo development. During this period, embryo cells undergo various physiological changes, which become evident by the deposition of storage materials, repression of germination and acquisition of desiccation tolerance. At the cotyledonary stage, energy requirements reach a maximum, suggesting the relevance of primary metabolite production, such as amino acids and fatty acids, whereas fermentation could constitute an alternative source of energy at the early stages (i.e., globular stage) of somatic embryo development [94]. This intense cellular reprogramming usually requires the removal of the auxin exogenously supplied in the induction step, and other PGRs assume critical roles, such as ethylene and ABA [95]. Exogenous ABA treatments are often used to increase somatic embryo maturation efficiency both in gymnosperms and angiosperms. In germinating Arabidopsis seeds, ABA induces the accumulation of miR159 in an ABI3-dependent fashion, and miR159 mediates cleavage of MYB101 and MYB33 transcripts in vitro and *in vivo*. In *L. leptolepis*, miR169 was described as involved in the maturation of the somatic embryo, by responding to ABA in addition to a possible role in the maintenance of embryogenic or non-embryogenic potential and the maturation of the somatic embryo [96].

In the model species Arabidopsis, the study of *dcl1* mutants evidence the crucial role of miRNAs, some of them not yet identified, as key inhibitors of the maturation program during embryogenesis, in part by repressing the master regulators *LEC2* and *FUSCA3* [97]. The trihelix TFs *ARABIDOPSIS 6B-INTERACTING PROTEIN1-LIKE1* (*ASIL1*) and *ASIL2* and the histone deacetylase HDA6/SIL1 were also identified as components that act downstream of miRNAs to repress the maturation program early in embryogenesis [97].

In several species, somatic embryo maturation normally occurs under high sucrose concentrations, not only within the conifers but also the angiosperms, which can be considered a strong stress for the plant cells. According to previous studies, high levels of miR397 and miR408 in developing somatic embryos can be related to such stress conditions [98].

In conifers, several miRNAs known to contribute to normal and synchronized development of somatic embryos were pointed out as potential biomarkers of SE, including miR156, miR159, miR166, miR167, miR168, miR171, miR397 and miR398 [24,96,99]. Nevertheless, there is evidence that a specific miRNA can participate at several stages of the SE process in different plant species. For instance, in maize, miR156 regulates embryogenic callus differentiation, in cotton it is necessary for globular embryo development whereas in *D. longan* and *L. leptolepis* it is important for cotyledonary embryo development [100]. In fact, miR156, one of the most conserved plant miRNAs families, is involved in the regulation of several crucial biological processes like fertility, juvenile to adult transition phase, root, leaf and fruit development and secondary metabolism, targeting *SQUAMOSA PROMOTOR BINDING PROTEIN LIKE* (*SPL*) transcription factor genes [101]. It has been considered a major regulator of early embryogenesis due to the highly repressed expression of SPLs in the eight-cell embryo in mutants [102]. The repression of SPL10 and SPL11 transcripts by miR156 leads to an inhibition of the expression of seed maturation genes, even if the embryo is not at a proper developmental stage for that to occur [50]. Additionally, miR156 has also been associated with SE induction in sweet orange (*C. sinensis*) [78] and csi-miR156a was the highest expressed miRNA during SE. The overexpression of this isomiR, leading to a respective downregulation of SPLs, was confirmed to play an important role in citrus SE by direct involvement in stress responses and hormone signaling pathways [103].

Beyond targeting PHB/PHV during the SE induction stage, miR165/166 has been associated with the repression of *WUSCHEL-RELATED HOMEOBOX5* (WOX5) TF, thus causing an impact in later stages of the SE process, given the role of WOX5 in the maintenance of the root apical meristem in Arabidopsis somatic embryos [77,104].

Other examples that reinforce the plant species dependency on the particular patterns of miRNAs expression during SE and embryo development, is the low expression of miR171, miR390 and miR398 before induction of embryo differentiation from *Oryza sativa* [105] and *L. leptolepis* [106] embryogenic calli, in contrast with their increase during the differentiation process in citrus species [78].

The dicot-specific miR403 has been described to accumulate during the late transition phase and persist in mature green embryos in Arabidopsis, which also suggests a role in embryo maturation [4]. Similarly, miR161, a miRNA only found in the Brassicaceae family, has shown accumulation during the transition phase, despite its low levels in mature embryos in Arabidopsis. Another Brassicaceae-specific, the miR824 targeting AGAMOUS-LIKE16, was reported to increase its levels during mid-embryogenesis, especially in the heart and early torpedo embryonic stages [4].

The miR482/miR2118 superfamily, characterized in *P. abies* [107], was found up-regulated in *P. pinaster* cotyledonary and mature somatic embryos when compared to zygotic embryos at the same stages of development [17]. This superfamily has been associated with the regulation of siRNAs biogenesis from the nucleotide-binding site-leucine-rich repeat (NB-LRR) gene family and triggering siRNA production in reproductive tissue, suggesting a dual function in gymnosperms. These functions were also individually retained in monocots and angiosperms [108,109,110,111]. The role that this family is playing in SE is still not clear; however, it was reported that NB-LRR genes play a role in hormonal responses to environmental stress [112], which could be in line with the conditions of reduced water availability applied for somatic embryo maturation SE protocols in these species [113].

## 5. Conclusions and Perspectives

The extensive studies on embryogenic cultures of seed plants within most taxonomic groups, but particularly on the model plant Arabidopsis, have to some extent disclosed the complexity of the interactions between miRNA-mediated regulation and hormone and stress signaling, of which the auxin-related pathways have been found to play a central role (Figure 1).

Despite the similarities between the genetic regulation of SE and ZE, recent RNAseq data of embryogenic cultures from both angiosperms (Arabidopsis) and gymnosperms (*P. pinaster*) reported that the transcriptomes of somatic embryos differ from the transcriptomes of zygotic embryo at equivalent stages of development, with somatic embryos bearing a resemblance to the gene expression pattern of germinating seeds [15,17]. This is not too surprising, given the artificial and mostly stressful conditions required for the differential gene expression that triggers SE. In fact, the analysis of Arabidopsis embryo coding transcriptomes revealed significant differences between zygotic and somatic embryos [15]. Several key regulators such as WOX2, WOX8, LEC1 and LEC2, were less abundant in somatic embryos, while other such as PLETHORA 1/2/3 were more abundant. A high correlation between the expression patterns of somatic embryos and germinating seeds was also found, suggesting that the timing of gene expression is altered in somatic embryos [15]. In fact, also the sncRNA profiles in maritime pine shows that although somatic and zygotic embryos express roughly the same repertoire of conserved and novel miRNAs, their expression profiles along development are different [17]. However, it should be pointed out that few reports have addressed the potential role of the zygotic embryo surrounding tissues (endosperm or megagametophyte), lacking in SE, in the miRNA-mediated regulation of the developing embryo. For instance, in *P. pinaster*, the megagametophytes show a rich non-coding transcriptome [17] which may have yet undiscovered roles in embryo development. The molecular networks that characterize the early vs. the later stages of SE [7,15] indicate that the most significant variations between distinct taxonomic groups are most likely to occur at the earliest stages, namely during induction of embryogenic competence. This emphasizes the challenge involved in the identification of the central components of the common molecular regulatory pathways that confer embryogenic capacity and, consequently, embryonic development from zygotic and somatic cells. Within these central components, miRNAs have been associated with the regulation of key TFs that modulate SE responses across all groups of land plants. Several questions, such as the miRNA roles, if any, at the initial embryo cell divisions in gymnosperms, following distinct patterns from those observed in angiosperms, remain unanswered.

In the past recent years, several studies have pointed out that most miRNAs are highly conserved throughout the plant kingdom, some of which were identified as crucial regulators of embryogenesis. Some miRNA families seem to be involved in the regulation of embryo development in both plant groups. However, other families, present throughout the embryogenic process, seem to be specific to each group (Table 1). According to specialized repositories such as miRbase (http://www.mirbase.org/) [114] (Accessed on 23 January 2021) and PmiREN (http://www.pmiren.com/) [115] (Accessed on 23 January 2021), some miRNA families are gymnosperm-specific, while others are angiosperm-specific, and within these, some are only found in monocots (such as *O. sativa, Z. mays* and *Triticum aestivum*) or in dicots (such as Arabidopsis). Based on a literature survey, Table 1 presents the different group-specific miRNA families reported as differentially expressed in tissues undergoing SE, although the roles of some of them, such as the Poaceae-specific miR5067 and the dicot-specific miR163, are yet to be determined. It is expected that the publication of more and larger miRNA datasets in the near future will significantly extend our understanding of their roles in embryogenesis. Furthermore, the release of new genome sequences and improved genome annotations will be important to address questions that still remain unanswered.

The adoption of novel approaches such as single cell transcriptomics should lead to a better resolution in the characterization of cell type-specific expression patterns, which are usually diluted in the tissue samples used in most of the reports published up to now. In any case, it will be always a major challenge to study SE due to the wide variety of different experimental conditions used to trigger embryo development from somatic cells. Slight variations in the concentration of applied PGRs may result in significant differences in miRNA expression [71], making it difficult to find common trends, and this is further hindered by genotype dependence. At last, it should be pointed out that currently, many studies are performed on the assumption that specific miRNA target transcripts, and their functions, are largely conserved between species. However, in many cases this may not be the case. The availability of bioinformatic tools with improved miRNA target prediction algorithms may help to address if the roles of relevant miRNAs are conserved, i.e., if their target transcripts show conserved functions. This could contribute to significant advances in the functional characterization of the large number of miRNAs identified in transcriptomic studies. Furthermore, in vivo validation of the miRNA-target interactions should be pursued, thus contributing to a better understanding of miRNA-mediated regulation of SE.

## Figures and Tables

**Figure 1 plants-10-00504-f001:**
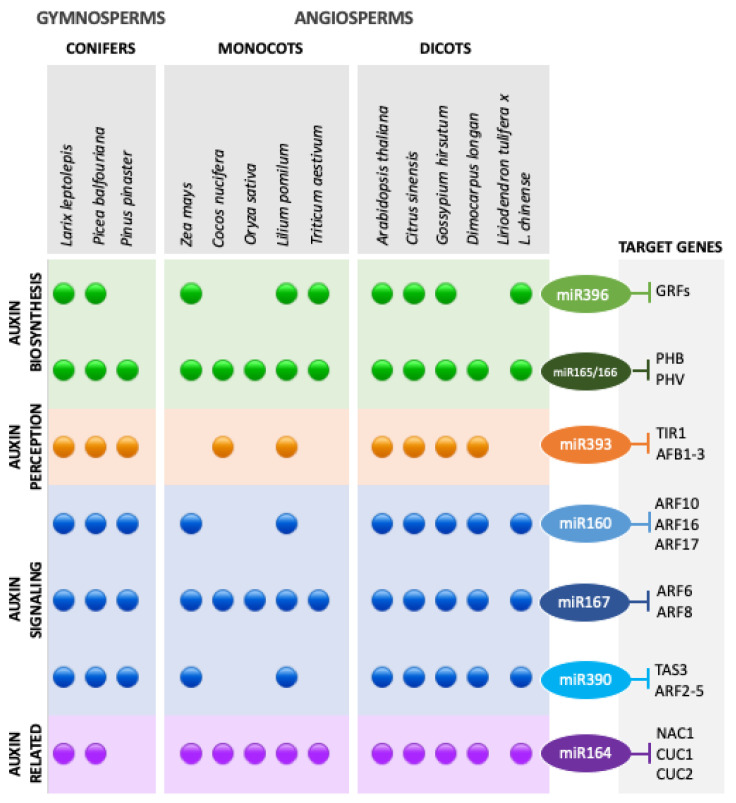
miRNAs reported as expressed (presence of dots) in SE transcriptomes of several seed plants within gymnosperms [17,70,71] and angiosperms groups, monocots [26,31,72,73,74,75,76] and dicots [32,42,52,66,77,78,79,80], involved in auxin biosynthesis (green), perception (orange), signaling (blue) and auxin-related (purple).

**Table 1 plants-10-00504-t001:** Group-specific miRNAs differentially expressed in embryogenesis.

		Somatic Embryogenesis (SE)-Related miRNA Family	Described or Putative Targets	(Putative) Role in Embryogenesis	References
**Gymnosperm-specific**		miR1314	Putative cellulose synthases (TIGR)	Embryo dormancy and germination	[28]
	**Conifer-** **specific**	miR946 miR947 miR951 miR1311	*unknown*	Embryo dormancy and germination	[28]
		miR950	NB-ARC		
		miR1312	GRF2, HB1		
		miR1313	LRK1		
		miR1315	receptor-like protein kinase	Embryogenic ability	[23]
		miR1316	LIP1, LIP2, TIR-NBS-LRR proteins	*Not* *determined yet*	[17,24,29]
		miR3699	*unknown*	*Not determined yet*	[17]
		miR3701	NBS-LRR proteins, cellulose synthase	*Not determined yet*	[17,29]
		miR3702 miR3704	*unknown*	*Not determined yet*	[24,29]
**Angiosperm-specific**		miR444	MIKC-type MADS-box	SE induction	[27]
		miR827	NLA and PHT5	Regulate auxin metabolism in early SE	[30]
	**Monocot-** **specific (Poaceae-** **specific)**	miR531	Wpk4 protein kinase	Embryogenic callus and embryo development	[26]
		miR1139	Myb1	Embryogenic callus development	[26]
		miR1878	NBS-LRR resistance protein-like		
		miR5049	Photosystem 1 subunit 5 Hydrolase, mitochondrial	Embryogenic callus and embryo development	[26]
		miR5067	Wpk4 protein kinase	*Not determined yet*	[31]
	**Dicot-** **specific**	miR158	SPINDLY	Gibberellic acid responses	[4]
		miR163	SAMT family members	*Not determined yet*	[32]
		miR403	AGO2, AGO3	Embryo maturation	[4]
		miR406	Spliceosomal proteins	Early embryogenesis	[33]
	**Brassicaceae-** **specific**	miR161	EMB2654, ARF	Embryo maturation	[4,34]
		miR824	AGAMOUS-LIKE16	Embryo maturation	[4]

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
