# Peer review of "Small Non-Coding RNAs at the Crossroads of Regulatory Pathways Controlling Somatic Embryogenesis in Seed Plants"

_plants, 2021, doi:10.3390/plants10030504_

Round 1
Reviewer 1 Report
The goal of this review article is to learn from studies on angiosperm development to better understand if and how conifer SE is regulated by sncRNAs. The review is comprehensive and as such demanding to read, but it is hard to see how it could be differently formatted - expression data from angiosperms and conifers are compared and discussed for different RNAs and tissues of expression. A figure showing different developmental stages and what RNAs are specifically expressed in angiosperms and conifers would be nice if possible, without being too busy.
Furthermore, specific questions are posed in the Introduction: “(i) Is initial zygotic/somatic 81 embryo development in gymnosperms regulated by miRNAs? (ii) Is regulation mediated 82 by the same miRNA families in both plant groups? (iii) Are the roles of these miRNAs 83 conserved, i.e., do their target transcripts show conserved functions?” - it seems pertinent to specifically respond to these questions in the conclusion. Alternatively, change the format to avoid the direct questions.
In general, as there are yet not many studies published on sncRNAs and also annotation of conifer sequences is far from satisfactory, some of the conclusions particularly on group specificity in the article will change in the future. This would be good to clearly acknowledge.
Section 3. on SE initiation generalises this process across species although the distinctions of angiosperms and conifer embryo developments are introduced in the Introduction as a main driver for the study. The SE initiation process is overall not well known and can arguably follow different paths in different plants and tissues, see Feher 2019 for review
https://www.frontiersin.org/articles/10.3389/fpls.2019.00536/full
Specifics:
Typo in Fig 1: gimnosperms
Reviewer 2 Report
- The manuscript is a review article on the role played by small non-coding RNAs in the control of somatic embryogenesis. It is well written and, although the language needs minor editing, it could be useful for international readers working on this subject. I have just some minor points to be considered by authors to improve the manuscript prior publication in this journal.
- In Arabidopsis thaliana (Arabidopsis) the characteristic different stages of embryo development are characterized ??????in detail and include the zygotic, globular, 54 heart, torpedo and cotyledonary (mature) embryo stages.
- A recent study showed that siRNAs generated from transposable elements (TEs), highly numerous in Arabidopsis embryos and typically 24-nt long, originate both, from euchromatic TEs being required not only during embryogenesis as a way of silencing TEs, and from heterochromatic TEs specifically REQUIRED???during embryogenesis, to help establish TE methylation de novo, which is then maintained during post-embryogenesis independently of siRNAs.
- It would be interesting if authors could include some information regarding relation between histone deacetylase inhibitors signaling and miRNAs, in as similar way as they did for strees signaling
- In addition, it would also be worthwile to include additional info regarding relations between miRNAs and somatic embryogenesis under in vivo conditions (nucellar embryos in angiosperms). In vivo asexual embryogenesis is a very adequate model to understand in vitro somatic embryogenesis.
- A general observation in angiosperms somatic embryogenesis in comparison to zygotic embryogenesis is that while gene expression in ZE occurs in a rather orderly manner, ca. genes coding for LEA proteins or order ABA-responsive genes are only expressed during the maturation phase, in SE, this type of genes are expressed since the early embryo stages ( globular) indicating the occurrence of strong dissimilarities in temporal gene expression between both type of processes. Could authors comment on it regarding miRNAs expression in ZE and SE.
Reviewer 3 Report
The authors did an extensive review, and commendable job to describe the role of sncRNA in embryogenic development of seed plants with a particular focus on model plant Arabidopsis. The review can be accepted for publication, provided the authors incorporated the following minor corrections.
Please rephrase line 39-42.
Line 52: change characteristic to distinct
Line 60: Replace the establishment of with establishing.
Please rephrase line 79-80
Line 82: what is the meaning of both plant groups? Gymnosperms and angiosperms?
Please rephrase line 100-105.
Please rephrase line 112-114.
Line 260-265: please split into two sentences.
Line 291: Evidences should be evidence.
Line 327: Related to not related with
Line 328-331: Please rephrase
Line 387: Also should be removed
Please rephrase line 426-427.
